# Environmental Impact Decision Support Tools for Horticulture Farming: Evaluation of GHG Calculators

**Arnis Dzalbs \*, Madara Bimbere, Jelena Pubule and Dagnija Blumberga**

Institute of Energy Systems and Environment, Riga Technical University, Āzenes iela 12/1, LV-1048 Riga, Latvia; madara.bimbere@rtu.lv (M.B.); jelena.pubule@rtu.lv (J.P.); dagnija.blumberga@rtu.lv (D.B.)
* Correspondence: arnis.dzalbs@rtu.lv

**Abstract:** Horticulture is essential in the European agricultural sector and fundamental for many EU member states. Decision Support Systems and Tools can play an essential role in a shift to result-based agriculture and evidence-based decision making, improving productivity and environmental performance of farming practices. Investigations have been conducted on horticulture crop and farming impact on the environment and Green House Gas emissions. Despite the availability of a broad spectrum of tools, the use of Decision Support Tools in agriculture in Europe could be much higher. This research aims to analyze and recommend environmental impact Decision Support Tools for small and medium-sized companies to approach, measure, and evaluate horticulture farming. The research methodology includes a systematic review, bibliometric analysis, Multicriteria Decision analysis, and a case study analysis. During the research, multiple tools, including calculators used for an impact assessment of the agricultural sector, were selected. After applying eligibility criteria, an in-depth analysis of eight of the most suitable calculators was performed. The results of the Multicriteria Decision analysis show that the Solagro Carbon Calculator, The Farm Carbon Calculator, and the Cool Farm Tool are recommended in the first place as Green House Gas calculators for farmers.

**Keywords:** agri-food systems; common agricultural policy; simplified environmental impact tools; climate neutrality; farm-scale; GHG calculator; horticulture





## 1. Introduction

The New European Union (EU) Common Agricultural Policy 2023–2027 is shaping the transition of the Agricultural sector to climate neutrality and sustainability. The future of agriculture is linked to modern, precise farming, leading to a fairer distribution of funding, depending on farming and emission reduction capacity. Decision Support Systems and Tools can play an essential role in a shift to result-based agriculture and evidence-based decision making, improving productivity and environmental performance of farming practices [1]. Different Decision Support Systems have been used in agriculture to help decision makers make optimal decisions regarding developing and managing a farm or a whole sector. Decision Support Systems have been used for decision making regarding water management [2,3], land use changes [4], overall farm management [5], better allocation of resources [6], and mitigation of climate change [7]. Ara et al., 2021 [8] analyzed different Decision Support Systems in irrigated agriculture, and Wong-Parodi et al., 2020 [9] described the role of Decision Support Tools in sustainability assessment.

Decision Support Tools allow for farmers, policymakers, and industry to make effective decisions, illustrating various outcomes from different practices. Rose et al., 2016 [1] listed 395 Decision Support Tools for United Kingdom farmers: software-based, paper-based, and apps.

Simplified sustainability Decision Support Tools have been delimited by Denef et al., 2012 [10] in Arulnathan et al., 2020 [11] as calculators, protocols, guidelines, and models.

Bibliometric analysis of Decision Support Tools in agriculture was used to explore a scientific interest field. Bibliometric analysis aims to show the state of the industry structure

and emerging trends, is applied to the broad scope and large datasets, and is quantitative and qualitative [12]. Figure 1 illustrates the result of the bibliometric analysis. Bibliometric analysis was used to analyze open-access articles from the Scopus and the Web of Science databases and resulted in visualization by VOSviewer software (VOSviewer version 1.6.19).

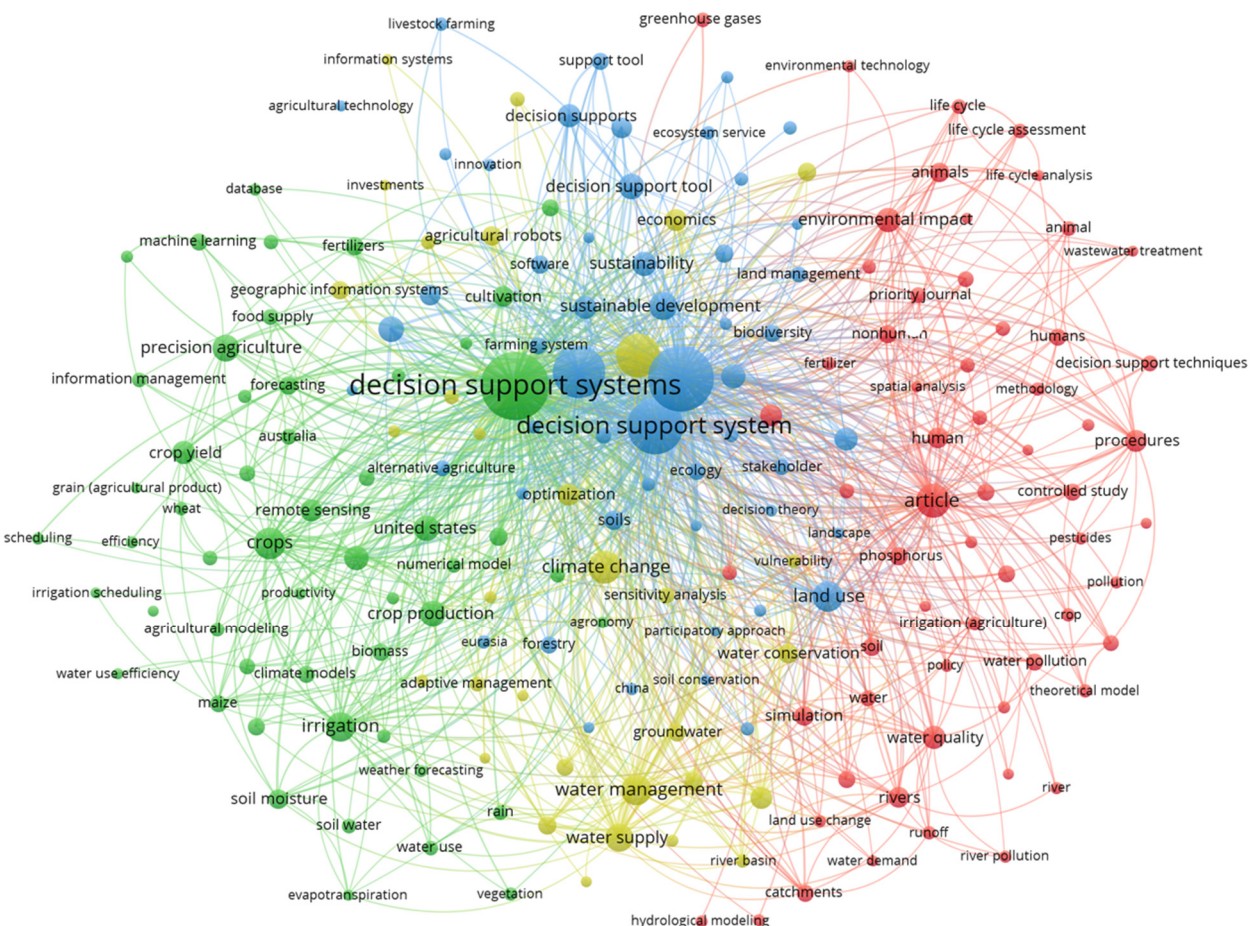

**Figure 1.** Keyword co-occurrence network for keywords "Decision Support Tools" and "Agriculture".

Articles from Scopus and Web or science databases were used for bibliometric analysis of open-access articles. Keywords "Decision Support Tools" and "Agriculture" were used as topics (search title, abstract, author keywords). When the used keywords appeared in the title, abstract, and author keywords, there were a total of 630 publications in Scopus and 1141 publications in Web of Science collected. Co-occurrences for all keywords equal to ten were used, and 198 of 5645 keywords met the threshold. All keywords were divided into four clusters (clusters are shown with different colors: blue, red, green, and yellow) with a more extensive cluster–Decision Support System. The results of a bibliometric analysis prove the topicality of the topic.

As shown in the bibliometric analysis, Decision Support Systems play an essential role in the sustainability assessment of the agriculture sector, and a broad range of applications and tools are used to analyze and improve the environmental performance of agricultural activities. A range of Decision Support Tools used in agriculture are described in the scientific literature. Olde et al., 2016 [13] focused on sustainability assessment tools at a farm level. As reported in Table 1 of the relevant publications, Whittaker et al., 2013 [14] evaluated carbon accounting tools for arable crops, and Colomb et al., 2013 [15] inspected greenhouse gas assessment (GHG) calculators for agriculture and forestry. In an article by Peter et al., 2017 [16], GHG emission calculators suitable for assessing energy crops were investigated. MacSween and Feliciano 2018 [17] summarized online GHG accounting tools

for Tropical climates. Renouf et al., 2018 [18] evaluated customized life cycle assessment tools for agriculture. Decision Support Tools can be analyzed depending on the scope of application, geographic scope, type of modeling, and level of assessment [11]; for example, Farm-level Decision Support Tools [11] or farm-level accounting models for a specific field—dairy cattle systems [19]. Thumba et al., 2022 focused on analyzing Decision Support Tools used to assess livestock farming [20] There are articles in the scientific literature about using specific tools for a particular agri-food sector, like wine [21]. Casson et al., 2023 [22], on an example of 79 simplified environmental impact tools used in the agri-food sector, categorized available solutions to support stakeholders in a distinction of the most convenient tool considering specific elements to consider.

**Table 1.** Decision Support Tool review studies.

| Publication | Field of Application | Tools Reviewed | References |
|---|---|---|---|
| Whittaker et al., 2013 | Arable crops in the United Kingdom | 11 | [14] |
| Colomb et al., 2013 | Landscape GHG assessment for agriculture and forestry | 18 | [15] |
| Arzoumanidis et al., 2014 | Application of simplified LCA in the wine sector | 4 | [21] |
| Olde et al., 2016 | Arable and livestock farm assessment in Denmark | 4 | [13] |
| Peter et al., 2017 | Carbon footprint calculators for energy crop cultivation | 18 | [16] |
| MacSween and Feliciano, 2018 | GHG Accounting Tools for Tropical Climate | 6 | [17] |
| Renouf et al., 2018 | Customized agricultural life cycle assessment tools | 14 | [18] |
| Arulnathan et al., 2020 | Farm-level Decision Support Tools | 19 | [11] |
| Vibart et al., 2021 | GHG emission models for dairy cattle systems | 14 | [19] |
| Thumba et al., 2022 | Decision Support Tools in livestock farming | 11 | [19] |
| Casson et al., 2023 | Simplified environmental impact tools for agri-food system | 79 | [22] |

Horticulture is essential in the European agricultural sector and is fundamental for many EU member states. Sector output comprises twenty-five percent of the total crop value in the EU and more than fourteen percent of the overall agricultural output value [23]. Horticulture contributes to the development of rural areas, encouraging sustainable food production and ensuring the achievement of the EU From Farm to Fork Strategy's targets [24,25]. European Plant Science Organization points to the need of the horticultural industry, dominated by small and medium-sized enterprises, for research-based Decision Support Tools available for farmers [25]. Carbon sequestration and developing "carbon farming" solutions suitable for horticulture is an actual topic in agriculture policy in the EU [26,27]. Carbon farming aims to mitigate climate change and shift to a green business model, like horticulture; therefore, a standard monitoring, reporting, and verification system is needed.

The calculations of GHG emissions from different horticulture practices using standardized tools is one of the possible solutions for monitoring agricultural activities and improving the ecological performance of a sector on the farm level. Multiple researches have been conducted about different horticulture crop and farming impact on the environment and GHG emissions. Soode et al., 2015 [28] measured the cradle-to-grave carbon footprints of strawberries. Schafer and Blanke, 2012 [29] analyzed the environmental impact of pumpkins in Germany. The authors refer to the limited information available about European horticultural product impact assessment and Decision Support Tools and, according to the results of bibliographic analysis, the United States of America, Australia, and Canada are the most analyzed countries.

Articles about perennial crops' carbon sequestration and the environmental performance of these cropping systems in Sweden [30], Spain [31], and Italy [32] analyze the effective strategies to combat climate change and refer to future investigation needs. GHG

emission calculations using Decision Support Tools in Iran [33–36] and Japan [37] are found in the scientific literature.

The availability and accessibility of such sustainability assessment tools have extended from only scientists to a broader range of stakeholders, including farmers, managers, and environmental officers [21]. While it is beneficial to account for more viewpoints, several challenges also arise, like data mismatching, performance, easiness of use, availability, and relevance.

Despite the availability of a broad spectrum of tools, using Decision Support Tools suitable for a farm level and focused on GHG emission estimation of the horticulture sector can be broader and more methodical. Although different Decision Support Tools for subcategories of agriculture have been evaluated, the contemplated literature review highlights the need for more guidelines for horticulture farmers on using sustainability assessment Decision Support Tools to estimate GHG emissions. The GHG calculator is an easy-to-use tool available for farmers, allowing the calculation and evaluation of the environmental performance of horticulture practices in Europe. A broad range of GHG calculators can be applied to different fields, including agriculture activities. A roadmap to choosing GHG calculators intended as a Decision Support Tool for European farmers in horticulture is needed.

Considering the previous conclusions discussed in the literature review, this research aims to analyze and recommend environmental impact Decision Support Tools—GHG calculators to approach, measure, and evaluate horticulture farming for small and medium-sized companies.

## 2. Methodology

The overall research methodology combines different scientific methods: systematic literature review, bibliometric analysis, Multicriteria Aecision analysis, and a case study analysis.

### 2.1. A Systematic Literature Review on Environmental Impact Decision Support Tools for Horticulture

A systematic review and meta-analysis were used to present the results of data from existing studies conducted on existing Decision Support Tools suitable for a farm level and focused on GHG emission estimation of the horticulture sector. Existing studies were evaluated during the systematic review process, which followed the Preferred Reporting Items for Systematic Reviews and Meta-Analysis PRISMA 2020 guidelines [38], and a statistical meta-analysis of results was conducted.

Bibliographic research to find literature matching the topic of the review was performed to contrive an introductory selection based on singular criteria to execute the selection process.

A systematic review aimed to review existing literature on agricultural Decision Support Tools. After the initial selection process, only the most relevant articles were used for review.

Inclusion criteria focused on simplified online environmental impact calculators for small and medium-sized horticulture farmers as a Decision Support Tool. Scopus and Web of Science databases were selected for review. Only open-access articles were included in the database search: 630 articles indexed in Scopus and 1141 articles indexed in Web of Science databases. In addition, the identification of new studies via other methods using the Google search engine was conducted. The first two hundred results proposed by the search engine were analyzed.

The keywords used were "Agriculture" and "Decision Support Tools". Eligibility criteria using five aspects were used to define tools for in-depth analysis: availability, scale, geographical scope, assessment unit, and status. The systematic review highlighted eight environmental impact Decision Support Tools—GHG calculators to approach, measure, and evaluate horticulture farming for small and medium-sized companies.

Before the screening, 560 articles were removed: duplicated records, records marked as ineligible by automation, and records removed for other reasons. In total, 611 articles were screened, and 500 articles were sought for retrieval. The eligibility criteria were selected for a review of Decision Support Tools—GHG calculators to approach, measure, and evaluate horticulture farming for small and medium-sized companies. Thirty-two articles were included in a systematic review of GHG calculators. The methodological framework of a systematic review is shown in Figure 2.

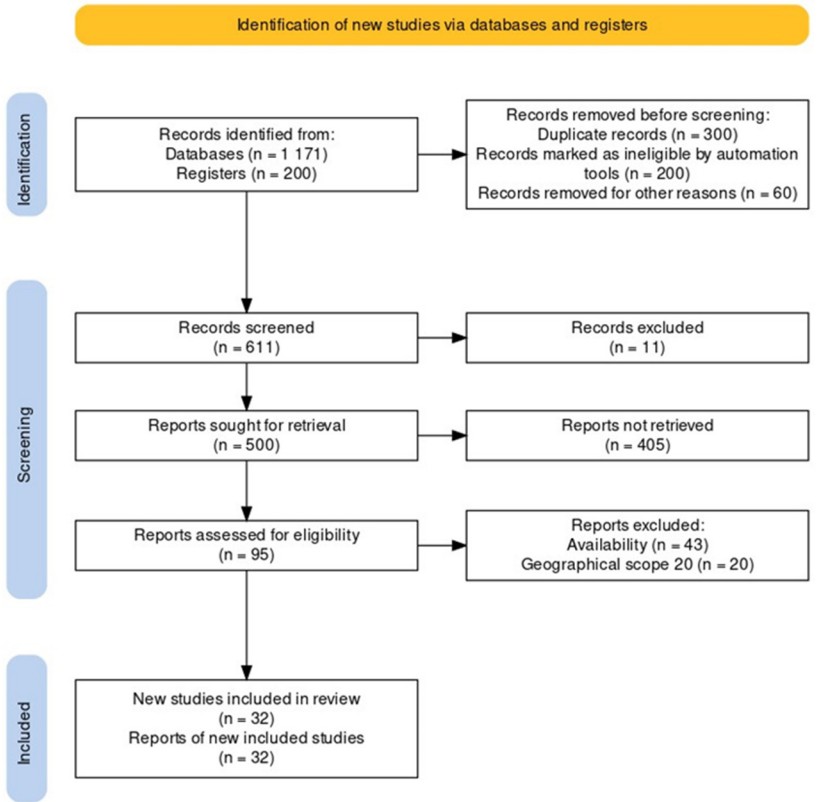

**Figure 2.** Flow diagram of systematic review.

### 2.2. Selection of Decision Support Tools—GHG Calculators

Based on the systematic literature review analysis results and the eligibility criteria selected for a review of Decision Support Tools—GHG calculators to approach, measure, and evaluate horticulture farming for small and medium-sized companies, eight GHG calculators were selected. To create a list of the most relevant tools, eligibility criteria were applied. Criteria considered in preliminary eligibility were (a) availability—free access, available after registration and available after purchase; (b) scale—farm-level, regional or global; (c) geographical scope—Europe, World, USA, Canada, and Australia; (d) assessment unit—crops, dairy, livestock, food, and beverage; (e) status—up to date, not updated, not working.

Considering the aim of the research and the needs of farmers, only calculators available for free or after registration were selected. Only the farm-level calculators were analyzed. Considering the need and overall scope, Europe as a primary scope and the world as a secondary scope were selected. Calculators for other regions, like the UK, Canada, Australia, and the USA, were excluded. After analysing the literature and applying eligibility analysis, eight calculators were selected as appropriate for an application on the farm level for horticulture practice analysis and evaluation: Agricultural Life Cycle Inventory Generator (ALCIG) [18,22], Common Agricultural Policy Regionalized Impact analysis (CAPRI) [39,40], The Farm Carbon Calculator [41], Integrated Management oPtions for Agricultural Climate Change miTigation (IMPACCT) [42], Model for integrative Life Cycle

Assessment in Agriculture (MILA) [43], Cool Farm Tool [44,45], Agricultural Resource Efficiency Calculator (AgreCalc) [20,46], and Solagro Carbon Calculator [47].

2.2.1. Bibliographic Analysis of Selected Indicators

Bibliometric analysis can be divided into two types: performance analysis, which shows the contribution of research, and scientific mapping, which shows the interrelationships, or linkages, of science. However, in addition to performance analysis and scientific mapping, network analysis complements bibliometric analysis: metrics, clustering, and visualization methods. The boundaries of the study were determined by the defined keywords used in the Web of Science comprehensive database, where publications were selected based on whether these keywords appear in the title, abstract, and author keywords.

In this bibliometric analysis, publication-related metrics were used because they show the topicality of the topic and the co-word analysis shows the relationships between words that frequently appear in publications and form clusters. VOSviewer software was used to visualize a biometric analysis network result. A Web of Science database was used for bibliometric analysis of the selected GHG calculators.

The folowing keywords were used: "Agricultural Life Cycle Inventory Generator" or "Common Agricultural Policy Regionalized Impact Analysis" or "The Farm Carbon Calculator" or "Integrated Management oPtions for Agricultural Climate Change miTigation" or "Model for Integrative Life Cycle Assessment in Agriculture" or "Cool Farm Tool" or "Agricultural Resource Efficiency Calculator" or "Solagro Carbon Calculator". When the used keywords appeared in the title, abstract, and author keywords, there were a total of 235 publications on the Web of Science. The results are illustrated in Figure 3 and show increased publications in recent years, commonly used in agriculture (29%) and environmental science research areas (24%).

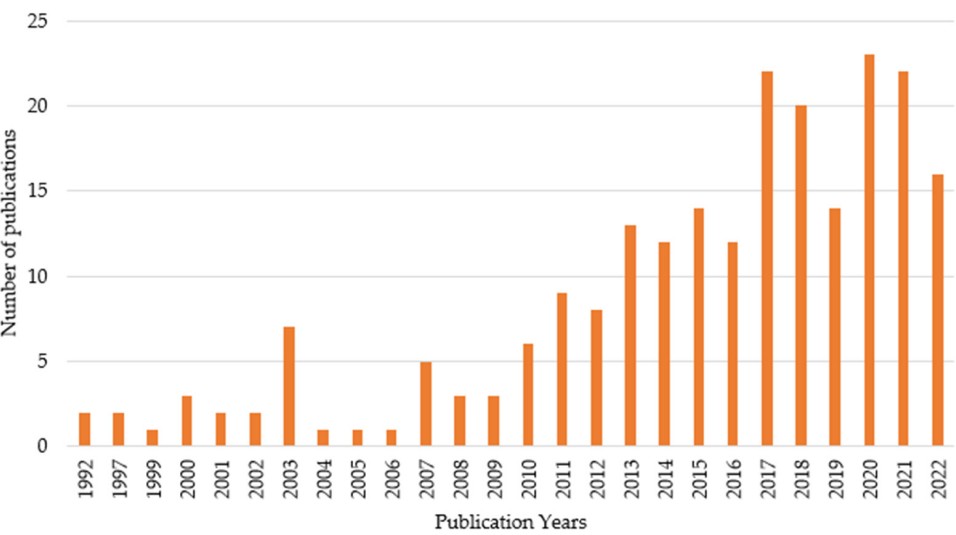

**Figure 3.** Bibliometric analysis of selected GHG calculators: publications.

In Figure 4, the keyword co-occurrence network is illustrated. Co-occurrence for all keywords equal to five was used. All keywords were divided into five clusters (clusters are shown with different colors: blue, red, green, yellow, and violet), with a more extensive cluster of agriculture (occurrence, 38; links, 61 and the second largest cluster—management (occurrence, 34; links, 61).

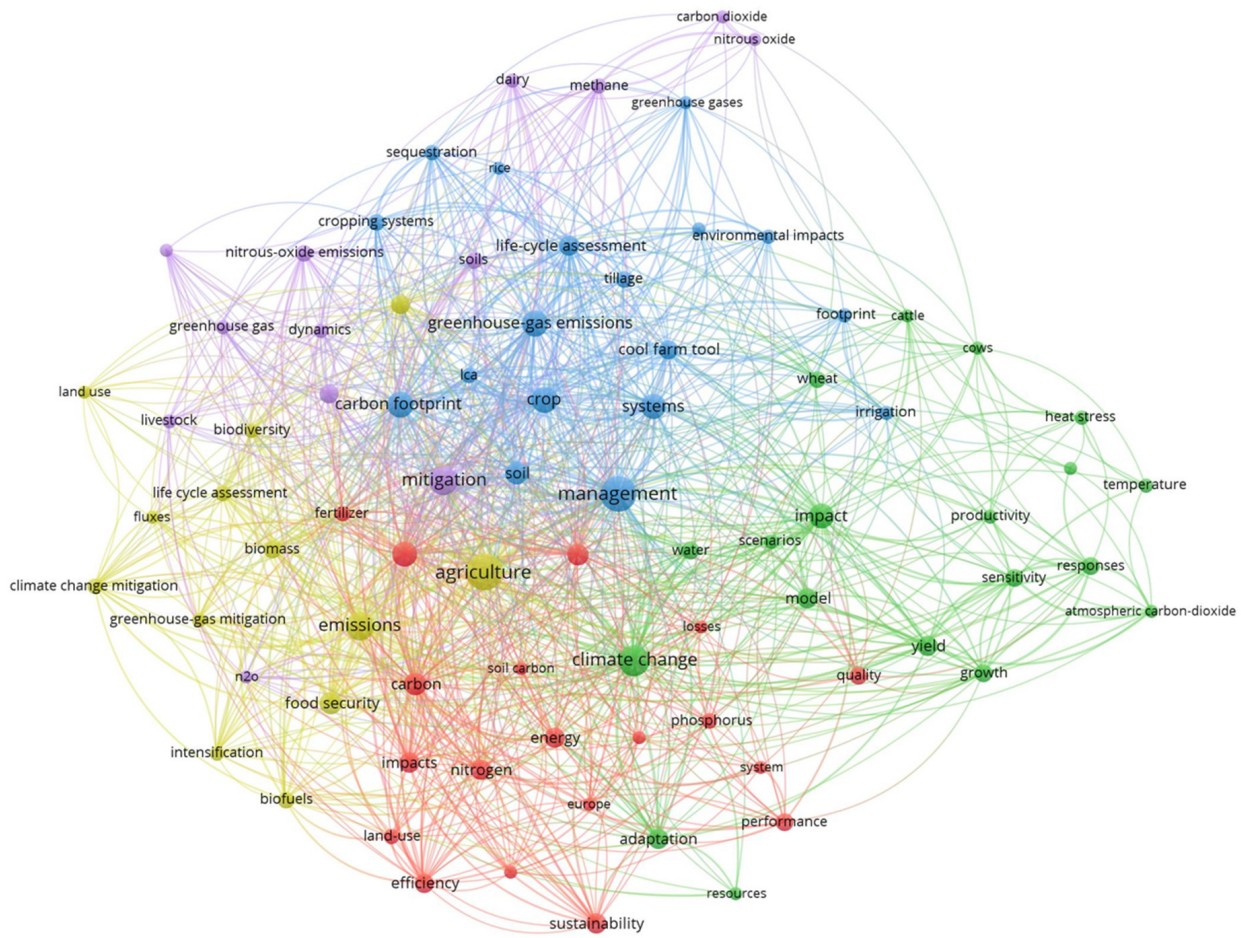

**Figure 4.** Keyword co-occurrence network of selected GHG calculators.

The results of bibliometric analysis prove the topicality and suitable application of selected Decision Support Tools—GHG indicators in horticulture farming for small and medium-sized companies.

2.2.2. In-Depth Analysis: Multicriteria Decision Analysis

During this stage, the in-depth analysis of eight calculators selected in the previous stages of methodology was conducted. Multicriteria Decision analysis is a process of sequent steps and processes to structure and formalize the decision-making process and allows the rank and prioritization of various selected options and, based on results, the making of recommendations on optimal options. According to the scientific literature, Multicriteria Decision analysis has been used in a sustainability assessment of agriculture [48] for the selection of mechanization patterns on farm levels [49] and crop production evaluation [50].

Selected indicators were used for an in-depth analysis and ranked. For the ranking, the Multicriteria Decision analysis, The Technique For Order of Preference by Similarity to Ideal Solution (TOPSIS), was used for the ranking of calculators for horticulture. This method evaluates the distance of alternatives to the ideal and anti-ideal point, and an option with the shortest distance to the ideal point is the best alternative.

The basic element of the TOPSIS analysis is a data matrix, where evaluation criteria are represented by $x_1, x_2, \ldots, x_j, \ldots, x_n$ (see Figure 5).

$$
\begin{array}{c}
\quad\quad x_1 \quad\quad x_2 \quad\ \cdots\ \quad x_j \quad\ \cdots\ \quad x_n \\
\begin{array}{c}
A_1 \\
A_2 \\
\vdots \\
A_i \\
\vdots \\
A_n
\end{array}
\left[
\begin{array}{cccccc}
x_{11}^k & x_{12}^k & \cdots & x_{1j}^k & \cdots & x_{1n}^k \\
x_{21}^k & x_{22}^k & \cdots & x_{2j}^k & \cdots & x_{2n}^k \\
\vdots & \vdots & \cdots & \vdots & \cdots & \vdots \\
x_{i1}^k & x_{i2}^k & \cdots & x_{ij}^k & \cdots & x_{im}^k \\
\vdots & \vdots & \cdots & \vdots & \cdots & \vdots \\
x_{n1}^k & x_{n2}^k & \cdots & x_{nj}^k & \cdots & x_{nm}^k
\end{array}
\right]
\end{array}
$$

**Figure 5.** TOPSIS decision-making matrix.

Different criteria used in the analysis have different dimensions. Value normalization is carried out to make these data comparable and, afterward, to rank alternatives in accordance with how closely they resemble the Positive Ideal solution. In this case, normalized values ($b_{ij}$) were obtained using linear normalization method [51].

$$b_{ij} = \frac{maxx_{ij} - x_{ij}}{maxx_{ij} - minx_{ij}}, \text{ if max } xij \text{ is preferable;} \tag{1}$$

$$b_{ij} = \frac{x_{ij} - minx_{ij}}{maxx_{ij} - minx_{ij}}, \text{ if min } xij \text{ is preferable.} \tag{2}$$

Normalized data were also arranged in a matrix and then weighted by multiplying them with the weights given to each of the criteria ($w_j$) (see Figure 6). There, {$A_1, A_2, \ldots, A_i, \ldots, A_m$} represents alternatives.

$$
\begin{array}{c}
\quad\quad w_1b_1 \quad\ w_2b_2 \quad\ \cdots\ \quad w_jb_j \quad\ \cdots\ \quad w_nb_n \\
\begin{array}{c}
A_1 \\
A_2 \\
\vdots \\
A_i \\
\vdots \\
A_n
\end{array}
\left[
\begin{array}{cccccc}
w_1b_{11}^k & w_2b_{12}^k & \cdots & w_jb_{1j}^k & \cdots & w_nb_{1n}^k \\
w_1b_{21}^k & w_2b_{22}^k & \cdots & w_jb_{2j}^k & \cdots & w_nb_{2n}^k \\
\vdots & \vdots & \cdots & \vdots & \cdots & \vdots \\
w_1b_{i1}^k & w_2b_{i2}^k & \cdots & w_jb_{ij}^k & \cdots & w_nb_{im}^k \\
\vdots & \vdots & \cdots & \vdots & \cdots & \vdots \\
w_1b_{n1}^k & w_2b_{n2}^k & \cdots & w_jb_{nj}^k & \cdots & w_nb_{nm}^k
\end{array}
\right]
\end{array}
$$

**Figure 6.** Normalized and weighted data matrix.

The next step of the TOPSIS analysis is to determine the Positive and Negative Ideal solutions.

The Positive Ideal solution:

$$A^+ = \text{Max}_i w_j b_{ij} \tag{3}$$

The Negative Ideal solution:

$$A^- = \min_i w_j b_{ij} \tag{4}$$

Separation from the Positive Ideal solution ($S^+$) is calculated by following formula:

$$S^+ = \sqrt{\sum_{j=1}^{n} \left(v_{ij} - v_j^+\right)^2}, \ i = 1, 2, \ldots, m. \tag{5}$$

Separation from the Negative Ideal solution:

$$S^- = \sqrt{\sum_{j=1}^{n} \left(v_{ij} - v_j^-\right)^2}, \ i = 1, 2, \ldots, m. \tag{6}$$

The final step is calculating alternatives of Relative Closeness to the Ideal Solution:

$$C_i^* = \frac{S_i^-}{\left(S_i^+ + S_i^-\right)}, \ i = 1, 2, \ldots, m. \tag{7}$$

The number obtained is in the range of [0;1] and shows the alternative rating. If $Ci^* = 1$, the alternative is equal to the Ideal Solution; if $C_i^* = 0$, it is the opposite of the Ideal Solution. The closer the rating is to 1, the better the alternative.

*2.3. Case Study Analysis*

Data from a horticulture farm in Latvia were used as a case study. The company has been engaged in sea buckthorn and quince cultivation since 2020. Data used to calculate GHG emissions are summarized in Table 2.

**Table 2.** Data used for calculations.

| Unit | Data |
|---|---|
| Total area | 12.5 ha |
| Area of planted sea buckthorn | 10 ha |
| Area of planted quinces | 0.5 ha |
| Unmanaged territory area | 2 ha |
| Number of sea buckthorn plants | 12,500 |
| Number of quinces plants | 1500 |
| Total yield | 31.5 t |
| Total sea buckthorn yield | 30 t |
| Total quinces yield | 1.5 t |
| Soil texture | loam |
| Soil drainage | very good |
| Soil management | no tillage |
| Plant residue management | removed from the field |
| pH | 5.7 |
| Amount of fertilizer—Kristalon Lilac (NPK 19-6-6+micro) [41] | 1400 kg; 140 kg per ha |
| Nitrogen (N) | 19%; 27 kg per ha |
| N (as ammonium N) | 16% |

**Table 2.** *Cont.*

| Unit | Data |
|---|---|
| N (as nitrate N) | 3% |
| Potassium oxide (K$_2$O) | 6%; 8.4 kg per ha |
| Phosphorus oxide (P$_2$O$_5$) | 6%; 8.4 kg per ha |
| Amount of fertilizer—Monopotassium phosphate [42] | 1000 kg; 100 kg per ha |
| Phosphorus pentoxide (P$_2$O$_5$) | 52%; 52 kg per ha |
| Potassium oxide (K$_2$O) | 34%; 34 kg per ha |
| Fuel consumption (tractor, mower, distribution truck) | 1200 L |
| Consumed electricity (drip irrigation system) | 3000 kWh |
| Irrigation amount | 2400 m$^3$ |
| Water source | farm storage pond |

## 3. Results: Prioritization of GHG Emissions Calculators for the Horticulture Sector

In total, eight calculators were selected for an in-depth analysis. All selected and analyzed environmental impact Decision Support Tools—GHG calculators are applicable for horticulture farming for small and medium-sized companies. The main characteristics of selected GHG calculators are summarized in Table 3.

**Table 3.** Farm-level GHG calculators suitable for the horticulture sector.

| Calculator | The Base of the Tool | Functional Unit | Sustainability Assessment | References |
|---|---|---|---|---|
| ALCIG | Excel, version 2310 | Surface, time | Single indicator | [18,22] |
| CAPRI | Software | Mass | Single indicator | [39,40] |
| The Farm Carbon Calculator | Web-based | Mass, surface | Single indicator | [41] |
| IMPACCT | Software | Mass, surface | Single indicator | [42] |
| MILA | Excel, version 2310 | Mass, surface, energy | Multi indicators | [43] |
| Cool Farm Tool | Web-based | Mass, surface | Multi indicators | [44,45] |
| AgreCalc | Web-based | Mass, surface | Single indicator | [20,46] |
| Solagro Carbon Calculator | Web-based | Mass | Single indicator | [47] |

As concluded from the literature review, a roadmap to choosing GHG calculators as a Decision Support Tool for European farmers in horticulture is needed. Considering the aim of this research—to analyze and recommend environmental impact Decision Support Tools—GHG calculators—prioritization of selected GHG calculators was conducted.

Indicators for prioritizing GHG emission calculators for evaluating the horticulture sector on the farm level were developed. Indicators were selected after examining the literature. The selected indicators are summarized in Table 4.

**Table 4.** Indicators used for a preference of GHG emission calculators.

| Dimension | Indicator | Designation of Indicator | Preferable Outcome |
|---|---|---|---|
| Technical | The base of the tool | I$_1$ | Max |
| Economic | Costs to farmers | I$_2$ | Min |
| Social | Convenience of use | I$_3$ | Max |
| | Recognition by farmers | I$_4$ | Max |
| Environmental | Transparency of methodology | I$_5$ | Max |

Values for technical, economic, and environmental dimensions were taken from the literature. Values for a social dimension were based on the authors' expertise and opinions. The equal weights of indicators were used in the analysis. Each indicator's weight was 0.2. The normalized and weighted matrix used for a TOPSIS analysis is shown in Table 5.

**Table 5.** Normalized and weighted TOPSIS matrix.

| Calculator | $I_1$ | $I_2$ | $I_3$ | $I_4$ | $I_5$ |
|---|---|---|---|---|---|
| ALCIG | 0.0295 | 0.0743 | 0.0645 | 0.0588 | 0.0555 |
| CAPRI | 0.0590 | 0.0743 | 0.0645 | 0.0294 | 0.0555 |
| The Farm Carbon Calculator | 0.0885 | 0.0743 | 0.0806 | 0.0882 | 0.0832 |
| IMPACCT | 0.0590 | 0.0743 | 0.0645 | 0.0588 | 0.0555 |
| MILA | 0.0295 | 0.0743 | 0.0322 | 0.0588 | 0.0555 |
| Cool Farm Tool | 0.0885 | 0.0743 | 0.0806 | 0.0882 | 0.0832 |
| AgreCalc | 0.0885 | 0.0371 | 0.0645 | 0.0882 | 0.0832 |
| Solagro Carbon Calculator | 0.0885 | 0.0743 | 0.0967 | 0.0735 | 0.0832 |

The results of the TOPSIS analysis allowed the identification of the three most advisable decision-making tools for use by horticultural farmers. The results of the TOPSIS analysis are shown in Figure 7.

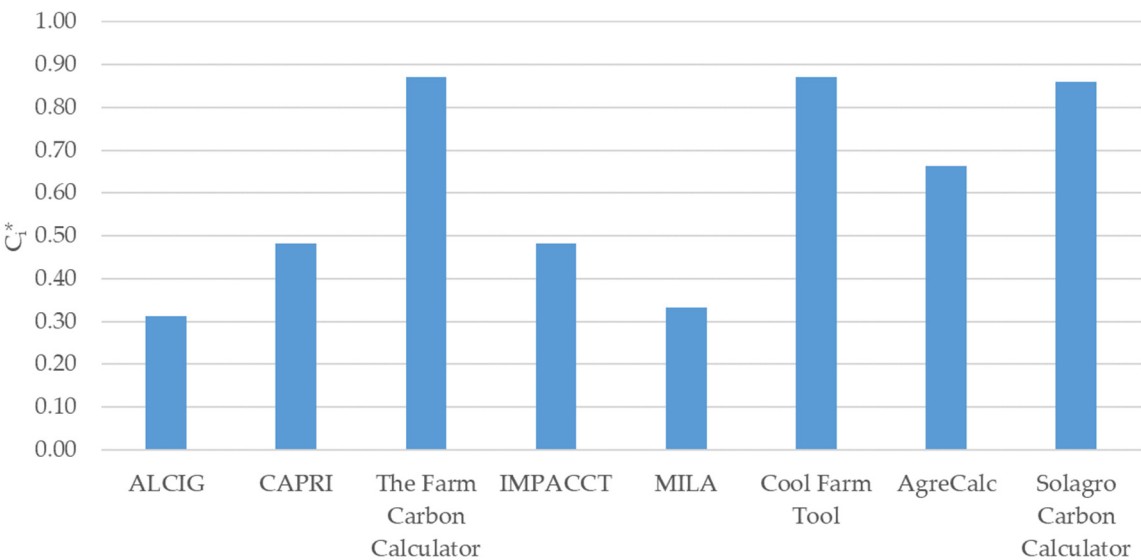

**Figure 7.** Comparison of ratings of GHG emission calculators.

All calculators were analyzed from technical, economic, social and environmental dimensions, using five indicators. The results show that the Solagro Carbon Calculator, The Farm Carbon Calculator, and Cool Farm have a higher rating and are recommended in the first place as GHG calculators for farmers. At the same time, it should be noted that all selected GHG calculators are applicable to approach, measure, and evaluate horticulture farming for small and medium-sized companies. For the rating of GHG emission calculators, equal weights of indicators were used. The equal weights were selected to minimize the subjectivity in the assessment. Therefore, sensitivity analysis was performed for transparency of a used approach, highlighting the dependence of relative closeness to the ideal solution on the weight (or importance) of criteria.

In this study, the weights of the criteria were equal to each other, and then each weight was changed separately to see how the overall results would be affected. According to the sensitivity graphs, Figure 8a–e criteria most depend on their weight in the case of cost to farmers, as variations in these criteria impact the ranking of final results. The results

show that the equal division of criteria weights is optimal, less subjective, and objectively compares calculators.

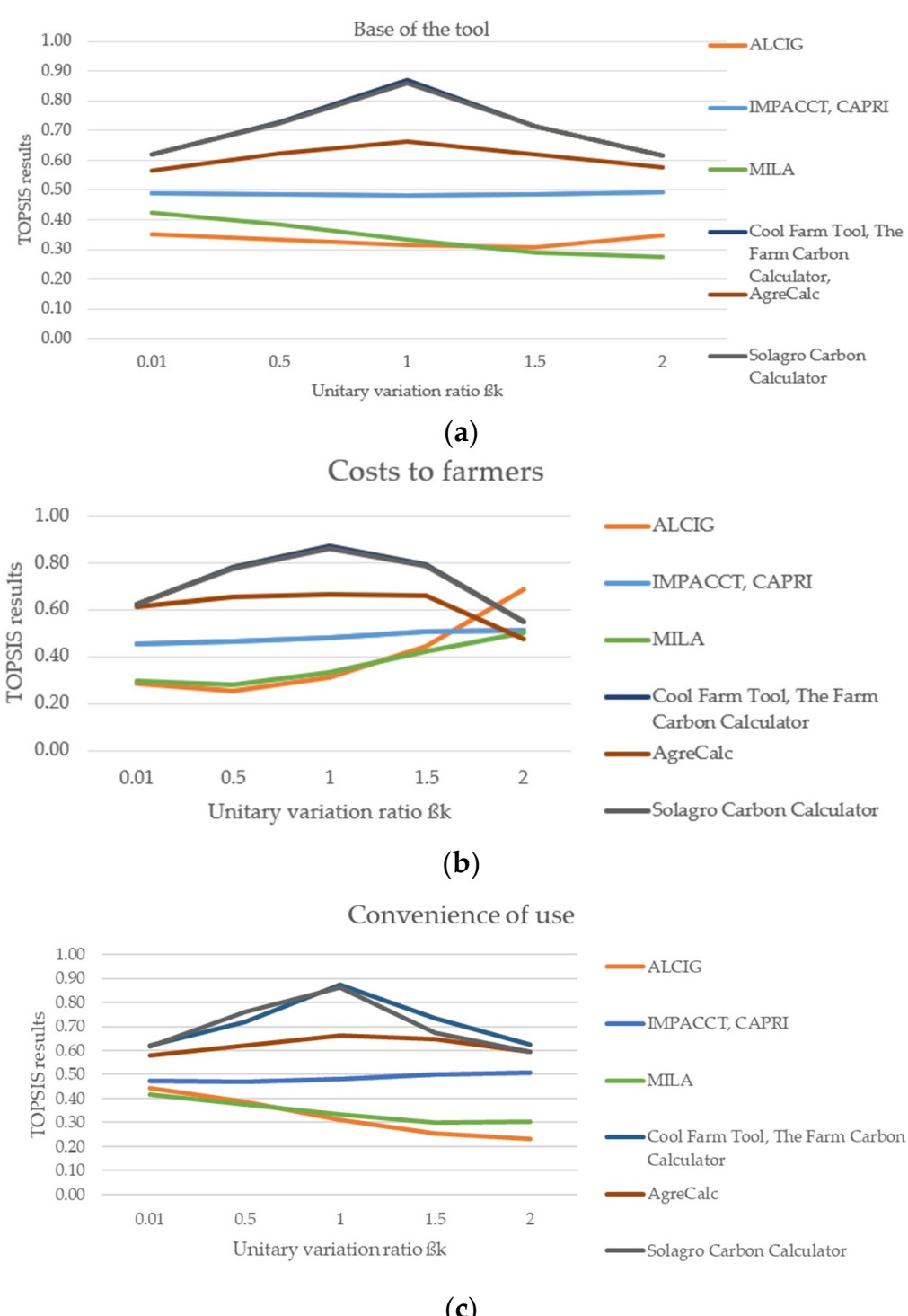

**Figure 8.** *Cont.*

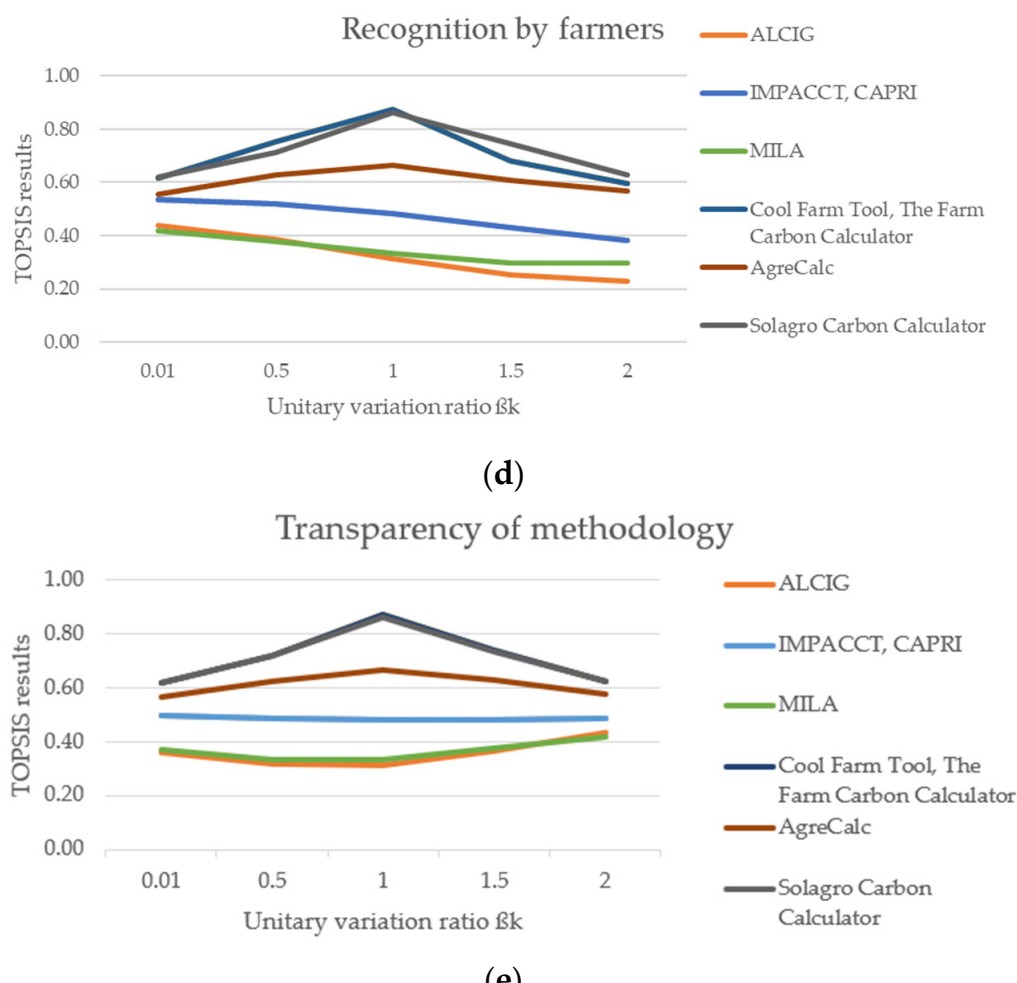

**Figure 8.** (**a**). Sensitivity analysis of the Base of the tool indicator. (**b**). Sensitivity analysis of the Cost to farmers indicator. (**c**). Sensitivity analysis of the Convenience of use indicator. (**d**). Sensitivity analysis of the Recognition by farmers indicator. (**e**). Sensitivity analysis of the Transparency of methodology indicator.

In future studies, a set of extended criteria should be brought forward to evaluate the effect on outcomes with respect to the first set of indicators selected.

As part of the review, GHG emission calculations were performed with three modeling tools with higher ratings (The Cool Farm Tool, The Farm Carbon Calculator, and Solagro Carbon Calculator) to examine these tools further and draw conclusions. The estimates were performed using information from the farm. For all three calculators, the same input data were used. Data used to calculate GHG emissions are summarized in Table 2.

Sea buckthorns have a drip irrigation system that delivers water and fertilizers, while quinces grow without watering. Information about the amount of fertilizers (Kristalon Lilac and Monopotassium phosphate) used on sea buckthorns was given in the unit of measure of kg. For The Cool Farm Tool and Solagro Carbon Calculator, it was necessary to convert the measurement units to kg per ha. The following equation for conversions was used:

$$a = \frac{b}{c},\qquad(8)$$

where

$a$—amount of fertilizer used, kg per ha;
$b$—amount of fertilizer used, kg;
$c$—total area of plants, ha.

The percentage distribution of fertilizer components is indicated on the product (Kristalon Lilac and Monopotassium phosphate) homepages, respectively [52,53]. Although the fertilizers contain additional nutrients, only N, $K_2O$, and $P_2O_5$ were considered in the work because adding additional nutrients to the NPK compound in the tools was impossible.

In The Cool Farm Tool and The Farm Carbon Calculator, fertilizer components were added in percentages. However, in the Solagro Carbon Calculator, it was necessary to calculate the amount (kg per ha) of each component. Equation (2) was used for calculation.

$$d = \frac{e}{100} \times a, \tag{9}$$

where

$d$—amount of fertilizer component, kg per ha;
$e$—amount of fertilizer component, %;
$a$—amount of fertilizer used, kg per ha.

The total fuel consumption of the tractor, mower, and distribution truck was given. The amount of cargo changed during each distribution time, so when calculating fuel consumption from distribution separately, the data were inaccurate; therefore, the consumption from distribution was included with the field operation machines and not calculated separately.

As an example, the total GHG emissions of the case study farm, according to calculations by the Solagro Carbon Calculator, are illustrated in Figure 9. The result chart created with the use of the Solagro Carbon Calculator shows the total emissions (tCO$_2$e) of the case study company, divided by sources (blue—machines and equipment, orange—process emissions, green—GHG emissions of energy used on the farm and purchased by thirds, purple—GHG emissions for other purchased inputs).

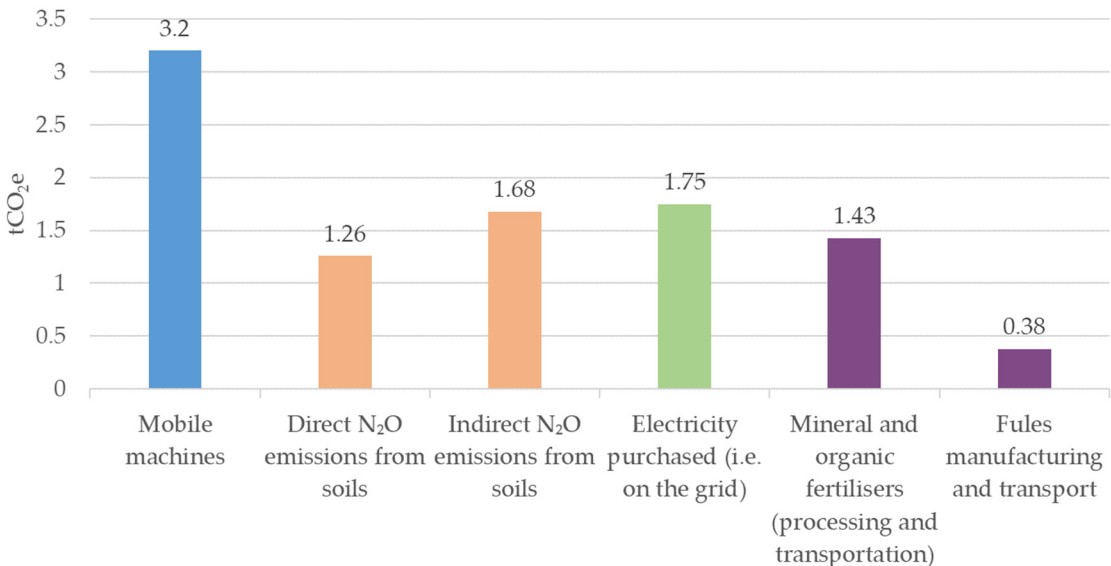

**Figure 9.** Results obtained by The Solagro Carbon Calculator.

All tools used data about crop area, harvest amount, soil type, fertilizer and fertilizer composition amounts, electricity consumed from irrigation and fuel by tractors and machines, and distribution. However, in The Cool Farm Tool and Solagro Carbon Calculator, it was also possible to enter data about crop residue management and pH concentration in the soil. As a result, emissions from fertilizer production were also considered; therefore, the results in The Cool Farm Tool appear higher.

## 4. Discussion and Conclusions

With increased focus on the result agriculture and sustainable development of a sector, farm-level evaluation is essential to support farmers in managing farms in a sustainable and competitive manner. Various applications and tools are used to analyze and improve the environmental performance of agricultural activities. Farmers widely use farm-level Decision Support Tools due to their characteristics: ease of use, ability to simplify the complexity of sustainability assessment, accessibility to farmers, and recognition by farmers and stakeholders. There is not and cannot be one universal Decision Support Tool. These tools vary due to focus, objectives, methods, and application differences. Therefore, a framework is needed to analyze and recommend environmental impact Decision Support Tools to approach, measure, and evaluate agriculture practices for small and medium-sized companies, including in horticulture.

During the research, a systematic review and bibliographic analysis were conducted, and different Decision Support Tools, including calculators used for an impact assessment of the agricultural sector, were investigated. GHG calculators play an essential role in the promotion of sustainable practices in agriculture and help to raise awareness about the need for a shift of agriculture practices towards climate neutrality. The need to access and monitor the environmental impacts of agriculture practices and services has resulted in the development of numerous GHG calculators.

From a broad range of simplified online environmental impact calculators for small and medium-sized horticulture farmers as a Decision Support Tool, only eight GHG calculators complied with the set of selection criteria: ALCIG; CAPRI, The Farm Carbon Calculator, IMPACCT, MILA, Cool Farm Tool, AgreCalcand, Solagro Carbon Calculator. Although all selected calculators are based on the IPCC guidelines, this does not provide a uniform approach or guarantee the same accuracy of results across all calculators. Each calculator addresses a different goal and varies in scoring, time investment, and data requirements. The IPCC guidelines provide a general framework, and models are built from a combination of methodology, and the level of detail differs; therefore, the results provided by different calculators may differ. Each calculator represents a different perspective on how the emissions are calculated; therefore, selecting the most suitable GHG calculator is essential: if carefully selected, it can result in correct findings and conclusions and provide sufficient information to assess environmental impacts. All selected and analyzed GHG calculators can be used to approach, measure, and evaluate horticulture farming for small and medium-sized companies. Farmers should work out the balance between efficiency and accuracy when deciding which calculator to use. To make the task easier for horticulture farmers, select GHG calculators were prioritized.

The most suitable GHG calculator tools should include a user-friendly platform for use, provide a comprehensive account of GHG emissions occurring on a farm level, be available in the public domain and free to use. Solagro Carbon Calculator, The Farm Carbon Calculator, and the Cool Farm Tool have a higher rating and are recommended in the first place as GHG calculators for horticulture farmers. These tools were acknowledged as the most relevant tools to gain insight into the sustainability performance of a horticulture farm.

This research is mainly addressed to European farmers in horticulture and can be used as a roadmap to (1) measure and compare horticulture farming, (2) evaluate Decision Support Tools available for horticulture, and (3) find an optimal GHG calculator on a farm-scale level. In future studies, the connection between the use of GHG calculators and subsequent changes in management practices should be investigated. Future effort in education and support of farmers is needed in using the outcomes of the calculations in decision making and improvement of farming practices to mitigate climate change and shift to a green business model.

**Author Contributions:** Methodology, A.D. and M.B.; Writing—original draft, M.B.; Writing—review and editing, J.P. and D.B.; Supervision, D.B.; Funding acquisition, A.D. All authors have read and agreed to the published version of the manuscript.

**Funding:** This work was supported by the European Social Fund within the Project No 8.2.2.0/20/I/008 "Strengthening of PhD students and academic personnel of Riga Technical University and BA School of Business and Finance in the strategic fields of specialization" of the Specific Objective 8.2.2 "To Strengthen Academic Staff of Higher Education Institutions in Strategic Specialization Areas" of the Operational Programme "Growth and Employment".

**Institutional Review Board Statement:** Not applicable.

**Data Availability Statement:** Publicly available datasets were analyzed in this study.

**Conflicts of Interest:** The authors declare no conflict of interest.

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
