# Peer review of "Environmental Impact Decision Support Tools for Horticulture Farming: Evaluation of GHG Calculators"

_agriculture, doi:10.3390/agriculture13122213_

Round 1
Reviewer 1 Report
Comments and Suggestions for Authors
Environmental impact decision support tools for horticulture farming: evaluation of GHG calculators
General comments: This is my first review of this manuscript. The topic is not particularly new, but it is still relevant and of increasing importance. The manuscript is generally well-written and engaging, however, the current draft is lacking transparency. In particular, the methodology is not clear. For example, further details are needed on the TOPSIS method, and the criteria used to rank the selected tools. It is not clear to me what the 'Ideal Solution' is. This needs much more explanation. The discussion and conclusions section also needs significant improvement (see specific comments below). There are a few other issues to address that I have outlined in my specific comments below.
Specific comments:
L12: "GHG", it is best to avoid abbreviations in the abstract.
L15: "conceal horticulture farming", this does not make sense - 'conceal' means to hide, is this right word?
L60: "the most convenient tool considering peculiar elements to consider", poor English/poorly written. Needs rewording… and not sure why the term peculiar is used?
L86: "easiness of use", would be better as "ease of use".
L94: "conceal horticulture farming", this does not make sense - 'conceal' means to hide, is this right word?
L102-106: This text is vague. More details are needed.
L111: I am struggling to see what value Figure 1 adds. Might be due to a lack of detail in the text. The authors should consider whether it is still need when they revise the text for the methodology (substantial revision is needed – see general comments).
L122-125: This text is vague. More details are needed.
L138-141: This is not clear / poorly written. Much more detail is needed to outline what the 'ideal Solution' is, and a clearer explanation of the 'normalisation' process is needed.
L158: Is Figure 2 needed? It might be better to describe this using text / bullet points. I don't think a diagram is needed and/or helps.
L189-193: It is difficult to understand how the chart in Figure 3 has been generated. This is due to the lack of a clear explanation of what the 'Ideal Solution' is, and thus how the 'Closeness to Ideal Solution' to the ideal solution has been calculated.
L189: "Closenes" should be Closeness.
L239: The text of some of the charts in Figure 4 is too small, making it difficult to read and interpret the charts.
L240-252: The 'discussion and conclusions' section is too short; indeed it does not include any significant discussion. I recommend the authors consider significantly expanding this section. For example, the authors emphasise the need to shift to 'result based agriculture' in their introduction; how do these tools support this evolution? It would also be good for the authors to reflect on their work to highlight its relative strengths and weaknesses, and with respect to the latter any limitations on the findings.
Comments on the Quality of English LanguageSee comments and suggestions for authors above.
Author Response
|
Reviewer 1
|
|
|
General comments: This is my first review of this manuscript. The topic is not particularly new, but it is still relevant and of increasing importance. The manuscript is generally well-written and engaging, however, the current draft is lacking transparency. In particular, the methodology is not clear. For example, further details are needed on the TOPSIS method, and the criteria used to rank the selected tools. It is not clear to me what the 'Ideal Solution' is. This needs much more explanation. The discussion and conclusions section also needs significant improvement (see specific comments below). There are a few other issues to address that I have outlined in my specific comments below.
We would like to express our gratitude for the valuable comments from reviewers. Thank you for the work! Correction in all sections of the manuscript has been done, the methodological framework improved as well as discussion and decision section.
|
|
|
Specific comments:
|
|
|
L12: "GHG", it is best to avoid abbreviations in the abstract.
|
Corrections made
|
|
L15: "conceal horticulture farming", this does not make sense - 'conceal' means to hide, is this right word? |
Corrections made
|
|
L60: "the most convenient tool considering peculiar elements to consider", poor English/poorly written. Needs rewording… and not sure why the term peculiar is used? |
Corrections made
|
|
L86: "easiness of use", would be better as "ease of use" |
Corrections made |
|
L94: "conceal horticulture farming", this does not make sense - 'conceal' means to hide, is this right word? |
Corrections made
|
|
L102-106: This text is vague. More details are needed. |
The information about the systematic review has been added and more explanation provided. |
|
L111: I am struggling to see what value Figure 1 adds. Might be due to a lack of detail in the text. The authors should consider whether it is still need when they revise the text for the methodology (substantial revision is needed – see general comments). |
The information about the systematic review has been added and more explanation provided. |
|
L122-125: This text is vague. More details are needed. |
The information about the systematic review has been added and more explanation provided. |
|
L138-141: This is not clear / poorly written. Much more detail is needed to outline what the 'ideal Solution' is, and a clearer explanation of the 'normalisation' process is needed. |
Description of a method added and explained. |
|
L158: Is Figure 2 needed? It might be better to describe this using text / bullet points. I don't think a diagram is needed and/or helps. |
Figure 2 modified to text. |
|
L189-193: It is difficult to understand how the chart in Figure 3 has been generated. This is due to the lack of a clear explanation of what the 'Ideal Solution' is, and thus how the 'Closeness to Ideal Solution' to the ideal solution has been calculated. |
Modifications made |
|
L189: "Closenes" should be Closeness. |
Modifications made |
|
L239: The text of some of the charts in Figure 4 is too small, making it difficult to read and interpret the charts. |
Figure modified |
|
L240-252: The 'discussion and conclusions' section is too short; indeed it does not include any significant discussion. I recommend the authors consider significantly expanding this section. For example, the authors emphasise the need to shift to 'result based agriculture' in their introduction; how do these tools support this evolution? It would also be good for the authors to reflect on their work to highlight its relative strengths and weaknesses, and with respect to the latter any limitations on the findings. |
Section extended and re-worked |
Reviewer 2 Report
Comments and Suggestions for Authors
This paper used systematic literature review, multi-criteria description analysis, and a case study to identify appropriate tools for GHG assessment tool for horticulture. They found three appropriate tools based on the methods and tried using these tools on a case study.
While the paper might be timely and appropriate, it lacks several important components needed to make it an acceptable publication. The literature review could benefit from drawing conclusions and stressing the importance and novelty of the study. The methods lack several details such as the review report according to PRISMA, the ways to get opinions from stakeholders, etc. The results were not explained and conclusions were not given beyond which tools were 'the best'. The discussion did not draw on previous literature or hint at the contribution to the field or implication at the industry.
These information should be added before the paper could by considered appropriate. More specific comments are attached.

Author Response
|
Reviewer 2 |
|
|
paper used systematic literature review, multi-criteria description analysis, and a case study to identify appropriate tools for GHG assessment tool for horticulture. They found three appropriate tools based on the methods and tried using these tools on a case study. While the paper might be timely and appropriate, it lacks several important components needed to make it an acceptable publication. The literature review could benefit from drawing conclusions and stressing the importance and novelty of the study. The methods lack several details such as the review report according to PRISMA, the ways to get opinions from stakeholders, etc. The results were not explained and conclusions were not given beyond which tools were 'the best'. The discussion did not draw on previous literature or hint at the contribution to the field or implication at the industry. These information should be added before the paper could by considered appropriate. More specific comments are attached.
We would like to express our gratitude for the valuable comments from reviewers. Thank you for the work! Correction in all sections of the manuscript has been done, the methodological framework improved as well as discussion and decision section.
|
|
|
L45 “The recent literature analysis provides different reviews of currently available decision support tools” What's the main idea of this paragraph? |
bibliographic analysis is added to show the idea of the paragraph and topicality.
The introduction section is rewritten in a more logical manner. |
|
L48 The literature is grouped together but the paragraph is not saying how it is relevant to the topic at hand.
|
The introduction section is rewritten in a more logical manner. |
|
L73 “Investigations have been conducted about different horticulture crops and farming's impact on the environment and GHG emissions” How is this important? |
Information about the role of monitoring and use of standardised tools is added. |
|
L87 “ Despite the availability of a broad spectrum of tools, the use of Decision support tools in agriculture in Europe could be much higher” This is a jump from a previous paragraph. Also, what are the evidence of the challenges and why do we need to find the best one? |
The introduction section has been improved. |
|
L 118 Please refer to PRISMA for more about how to report the systematic review methods. |
Reference to Prisma guidance for a systematic review added. |
|
L 128 How are these eight selected? |
Information about the selection process via systematic and bibliographic analysis added |
|
L173 Were these mentioned in the methods? |
Corrections made |
|
L179 How are these collected and were the methods approved by the REC/IRB? I didn't see this in the method section. |
The approprite description of a process added |
|
L 186 The results of the TOPSIS analysis allowed the identification of the three most advis- 186 able decision-making tools for use by horticultural farmers.
|
Corrections made |
|
L233 What do these mean? |
Explanation of results given. |
|
L243 Discussion should include key findings and what they mean, contribution to the field, implications, and future study. This is barely a conclusion. None of the important parts appear here. |
Discussion and Conclusion sections have been improved. |
Round 2
Reviewer 1 Report
Comments and Suggestions for Authors
Environmental impact decision support tools for horticulture farming: evaluation of GHG calculators
General comments: The methodology for ranking the tools is still lacking transparency. More details on the TOPSIS method have been provided, but the method is still lacking details on the criteria that are used within the multi-criteria analysis. Table 3 lists the criteria used, i.e. Base of the tool; Costs to farmers; Convenience of use; Recognition by farmers; and Transparency of methodology. But there are no details about these criteria. For example, how were each of these measured/assessed? What scales and/or scoring techniques are used to make a judgement of what is ideal or not ideal? This detail needed to understand the basis for judging what is the ideal solution. The discussion and conclusion section is still quite weak. Issues raised in my previous review have not been adequately addressed. Indeed, this section looks rushed and not thought through. I suggest the authors spend a bit more time to draft a more thorough discussion and neater conclusion for their work.
Comments on the Quality of English LanguageNone
Author Response
|
General comments: This is my first review of this manuscript. The topic is not particularly new, but it is still relevant and of increasing importance. The manuscript is generally well-written and engaging, however, the current draft is lacking transparency. In particular, the methodology is not clear. For example, further details are needed on the TOPSIS method, and the criteria used to rank the selected tools. It is not clear to me what the 'Ideal Solution' is. This needs much more explanation. The discussion and conclusions section also needs significant improvement (see specific comments below). There are a few other issues to address that I have outlined in my specific comments below. We want to express our gratitude for the valuable comments from a reviewer. Correction in all manuscript sections has been done, the methodological framework improved, and the discussion and decision section has been improved.
In particular, the methodology is not clear. For example, further details are needed on the TOPSIS method, and the criteria used to rank the selected tools. It is not clear to me what the 'Ideal Solution' is. This needs much more explanation.
The methodology part has been supplemented with an explanation of the TOPSIS analysis and how the "best" tools were evaluated. L282-317 L-388-415. and
The discussion and conclusions section also needs significant improvement (see specific comments below).
The discussion and conclusions section was modified, and the conclusion rased from the research was added. L 474-535.
|
|
|
L12: "GHG", it is best to avoid abbreviations in the abstract.
|
Corrections made L12 L23
|
|
L15: "conceal horticulture farming", this does not make sense - 'conceal' means to hide, is this right word? |
Corrections made L16, 133, 140, 172, 179, 206, 393
|
|
L60: "the most convenient tool considering peculiar elements to consider", poor English/poorly written. Needs rewording… and not sure why the term peculiar is used? |
Corrections made L90 |
|
L86: "easiness of use", would be better as "ease of use" |
Corrections made L132 |
|
L94: "conceal horticulture farming", this does not make sense - 'conceal' means to hide, is this right word? |
Corrections made L140 |
|
L102-106: This text is vague. More details are needed. |
The methodological part has been improved, and precise information about the systematic review has been added. The precise information about the systematic review was added to the manuscript section. L149-182.
|
|
L111: I am struggling to see what value Figure 1 adds. Might be due to a lack of detail in the text. The authors should consider whether it is still need when they revise the text for the methodology (substantial revision is needed – see general comments). |
The information about the systematic review has been added and more explanation provided. Fig. 1 according to PRISMA guidelines, flow diagramme has been used. L149-182. |
|
L122-125: This text is vague. More details are needed. |
|
|
L138-141: This is not clear / poorly written. Much more detail is needed to outline what the 'ideal Solution' is, and a clearer explanation of the 'normalisation' process is needed. |
The methodology part has been supplemented with an explanation of the TOPSIS analysis and the “ideal solution” explaned and information about the normalisation process added. L282-317 L-388-415.
|
|
L158: Is Figure 2 needed? It might be better to describe this using text / bullet points. I don't think a diagram is needed and/or helps. |
Figure 2 modified to text. L213-217 |
|
L189-193: It is difficult to understand how the chart in Figure 3 has been generated. This is due to the lack of a clear explanation of what the 'Ideal Solution' is, and thus how the 'Closeness to Ideal Solution' to the ideal solution has been calculated. |
Modifications made in methodological part (L149-181) and in results part (L381-415) |
|
L189: "Closenes" should be Closeness. |
Modifications made L319 |
|
L239: The text of some of the charts in Figure 4 is too small, making it difficult to read and interpret the charts. |
Figure modified and resluts of only one calculator left in a figure. L455-475 |
|
L240-252: The 'discussion and conclusions' section is too short; indeed it does not include any significant discussion. I recommend the authors consider significantly expanding this section. For example, the authors emphasise the need to shift to 'result based agriculture' in their introduction; how do these tools support this evolution? It would also be good for the authors to reflect on their work to highlight its relative strengths and weaknesses, and with respect to the latter any limitations on the findings. |
The discussion and conclusions section has been revised and expanded. Main conclusions and limitations added and highlighted next steps and limitations of the research. L489-L542 |

Reviewer 2 Report
Comments and Suggestions for Authors
The authors adequately addressed all my comments.
Author Response
|
Paper used systematic literature review, multi-criteria description analysis, and a case study to identify appropriate tools for GHG assessment tool for horticulture. They found three appropriate tools based on the methods and tried using these tools on a case study. While the paper might be timely and appropriate, it lacks several important components needed to make it an acceptable publication. The literature review could benefit from drawing conclusions and stressing the importance and novelty of the study. The methods lack several details such as the review report according to PRISMA, the ways to get opinions from stakeholders, etc. The results were not explained and conclusions were not given beyond which tools were 'the best'. The discussion did not draw on previous literature or hint at the contribution to the field or implication at the industry. These information should be added before the paper could by considered appropriate. More specific comments are attached. We want to express our gratitude for the valuable comments from a reviewer. Correction in all manuscript sections has been done, the methodological framework improved, and the discussion and decision section has been improved.
The literature review could benefit from drawing conclusions and stressing the importance and novelty of the study.
The literature part has been improved and supplemented by conclusions and adding importance and underlining the study's novelty. L 49-73 L102-108 L113-115 L126-137. The methods lack several details such as the review report according to PRISMA, the ways to get opinions from stakeholders, etc. The methodological part has been improved, and precise information about the systematic review according to PRISMA has been added. The precise information about the systematic review was added to the manuscript section. L149-182. The results were not explained and conclusions were not given beyond which tools were 'the best'. The methodology part has been supplemented with an explanation of the TOPSIS analysis and how the "best" tools were evaluated. L282-317 L-388-415. The discussion did not draw on previous literature or hint at the contribution to the field or implication at the industry. The discussion and conclusions section was modified, and the conclusion rased from the research was added. L 474-535. |
|
|
L45 “The recent literature analysis provides different reviews of currently available decision support tools” What's the main idea of this paragraph? |
Bibliographic analysis is added to show the idea of the paragraph and topicality. L49-73.
The introduction section is rewritten in a more logical manner. |
|
L48 The literature is grouped together but the paragraph is not saying how it is relevant to the topic at hand. |
The introduction section is rewritten in a more logical manner. L49-73.
|
|
L73 “Investigations have been conducted about different horticulture crops and farming's impact on the environment and GHG emissions” How is this important? |
Information about the role of monitoring and use of standardised tools is added. L102-108. |
|
L87 “ Despite the availability of a broad spectrum of tools, the use of Decision support tools in agriculture in Europe could be much higher” This is a jump from a previous paragraph. Also, what are the evidence of the challenges and why do we need to find the best one? |
The introduction Section has been improved. L126-137. |
|
L 118 Please refer to PRISMA for more about how to report the systematic review methods. |
Reference to Prisma guidance for a systematic review added. L149-182. |
|
L 128 How are these eight selected? |
Information about the selection process via systematic and bibliographic analysis added L202-262. |
|
L173 Were these mentioned in the methods? |
Corrections made, in a first version of the manuscript unprecised information was added. Indicators were selected after examining the literature.
|
|
L179 How are these collected and were the methods approved by the REC/IRB? I didn't see this in the method section. |
The approprite description of a process added
Corrections made, in a first version of the manuscript unprecised information was added. L370-372. |
|
L 186 The results of the TOPSIS analysis allowed the identification of the three most advis- 186 able decision-making tools for use by horticultural farmers. |
This part of the results have been revised and a clarification given. L388-415. |
|
L233 What do these mean? |
Explanation of results given. L455-474. |
|
L243 Discussion should include key findings and what they mean, contribution to the field, implications, and future study. This is barely a conclusion. None of the important parts appear here. |
Discussion and Conclusion sections have been improved. L477-535 |
